# Solid-State Foaming of Acrylonitrile-Butadiene-Styrene/Recycled Polyethylene Terephthalate Using Carbon Dioxide as a Blowing Agent

**DOI:** 10.3390/polym11020291

**Published:** 2019-02-09

**Authors:** Dong Eui Kwon, Byung Kyu Park, Youn-Woo Lee

**Affiliations:** 1School of Chemical and Biological Engineering and Institute of Chemical Processes, Seoul National University, 1 Gwananak-ro, Gwanak-gu, Seoul 151-744, Korea; sot5656@snu.ac.kr; 2Research Institute of Advanced Materials, Seoul National University, 1 Gwananak-ro, Gwanak-gu, Seoul 151-744, Korea

**Keywords:** acrylontrile-butadiene-styrene, recycled polyethylene terephthalate, solid-state foaming, carbon dioxide, microcellular foam, cell size distribution

## Abstract

In this study, a single paragraph of acrylonitrile-butadiene-styrene (ABS)/recycled polyethylene terephthalate (R-PET) polymeric foams is prepared using CO_2_ as a blowing agent. First, the sorption kinetics of subcritical and supercritical CO_2_ are first studied at saturation temperatures from −20 to 40 °C and a pressure of 10 MPa, in order to estimate the diffusion coefficient and the sorption amount. As the sorption temperature increases, the diffusion coefficient of CO_2_ increases while the sorption amount decreases. Then, a series of two-step solid-state foaming experiments are conducted. In this process, a specimen is saturated with liquid CO_2_ and foamed by dipping the sample in a high-temperature medium at 60 to 120 °C. The effects of foaming temperature and depressurization rate on the morphology and structure of ABS/R-PET microcellular foams are examined. The mean cell size and the variation of the cell size distribution increases as the foaming temperature and the depressurization rate increases.

## 1. Introduction

Plastics, whether synthetic or semi-synthetic polymeric compounds, play a significant role as building blocks in supporting the modern world. They are cheap, malleable, and water-resistant. Hence, they constitute a variety of products from daily necessities to building elements. Their versatility is preferred over traditional materials including woods, rocks, ceramics, and metals. However, the predominance of plastic products has also engendered environmental hazards. Since plastics degrade slowly in nature, a large amount of plastics that are disposed or landfilled are expected to disturb the food chain in the ecosystem. Plastics already make up 50%–80% of debris in marine areas [1] and continuously increase in proportion. The pileup of plastics in natural environments led us to seek ways to improve the recycling rate of these materials. 

Polyethylene terephthalate (PET), one of the most common polymers that is widely used, is a representative example. It is predominantly used as a packaging material, a building element, and in polyester fibers, but it is both expensive and difficult to degrade it through a biodegradation process. Hence, the recycling of post-consumer PET has been regarded as an economically feasible method to upcycle the polymeric material. Since recycled polymers deteriorate easily by contamination [2], adding recycled polymer as a constituent of a polymer blend is regarded as one of the practical solutions to utilize recycled PET (R-PET). R-PET has been added into a variety of polymer blends including polypropylene (PP) [3], polyolefin (POE) [4], polycarbonate (PC) [5], and acrylonitrile-butadiene-styrene (ABS) terpolymer [6,7]. Among them, the ABS/R-PET polymer blend is expected to be applied for a variety of uses since both materials show excellent thermal insulation as well as toughness. 

Meanwhile, the preparation of microcellular foams of ABS and PET by a supercritical or a gas foaming process have been independently investigated. Nawaby et al. studied the preparation of nanocellular foams and the CO_2_-induced retrograde vitrification of ABS [8]. Since this retrograde behavior facilitates the diffusion of CO_2_ and the formation of a cellular structure by its plasticization effect, ABS microcellular foams were easily obtained from the solid-state foam preparation process. Forest et al. manufactured polymeric foams of three types of ABS terpolymer, using a CO_2_ foaming process. Their foam structures had an average cell diameter that ranged from 500 nm to 2 µm and the cell number densities from 10^9^ cells/cm^3^ to 10^13^ cells/cm^3^ [9]. Yoon et al. also prepared the ABS foams using liquid CO_2_ as a blowing agent. They produced nanocellular and microcellular structures, which had a network of closed cells [10]. There have also been several studies that were performed in order to fabricate microcellular PET foams. However, harsh conditions were required to obtain the microcellular foams since PET crystallizes when it is exposed to high-pressure CO_2_. Li et al. investigated the crystallization behavior of PET under high-pressure CO_2_, and the cellular structure formation in crystalline and amorphous domains. The preparation of microcellular PET foam required a long saturation time, up to 15 days, and a high foaming temperature (235 °C). They also discovered that the cellular structures in the crystalline and amorphous regions had different foam structures [11]. Consequently, the preparation of neat ABS and PET microcellular foams has been studied, but the ABS/R-PET foam has not been prepared using subcritical or supercritical CO_2_ as a blowing agent, despite its environmental significance. In addition, since Zhao et al. reported that the PET blend with 70 wt % ABS inhibited the crystallization of PET [12], fabricating the microcellular foams of the same composition would be favorable because ABS has the advantage in terms of its foaming process and the blend form could modify the disadvantage of producing PET crystal.

Therefore, this work aims to prepare the ABS/R-PET microcellular foam, using CO_2_ as a blowing agent. First, the sorption kinetics of CO_2_ in the polymeric blend is measured. Then, a series of batch foaming experiments are carried out and the influence of process conditions on the morphology and the cell size distribution of the foams are explored.

## 2. Materials and Methods 

### 2.1. Materials

We purchased ABS/R-PET pellets (GC-0700) from Lotte Advanced Materials (Uiwang, Republic of Korea), which consisted of approximately 70 wt % of ABS and 30 wt % of recycled PET. The specific gravity of the raw material was 1.09, and the melting index (MI) of the material was 40 g/10 min. The pellets were molded into a disc by placing approximately 0.3 g of the pellets on the hot press machine at 250 °C and pressing them. The diameter and the height of each disc were 20 and 1 mm, respectively. CO_2_ (99.95%, Hyupshin Gas Industry, Seoul, Republic of Korea) was used as a blowing agent. All materials were used without additional purification.

### 2.2. Differential Scanning Calorimetry

Glass transition temperatures ($T_{g}$) and the crystallization of the ABS/R-PET disc were measured using the differential scanning calorimetry (DSC, Q100, TA instruments, New Castle, DE, USA). 5 mg of the sample was heated to 300 °C at a rate of 10 °C/min under a nitrogen environment. The $T_{g}$ of each PET and ABS in the blend was measured as 75.63 and 99.71 °C, respectively. 

### 2.3. Sorption Kinetics

A gravimetric method [13,14,15] was used to measure the sorption kinetics and the amount of CO_2_ absorbed in the polymeric disc. An ABS/R-PET disc was first weighed on a scale (EPG214, OHAUS, Pine Brook, NJ, USA) and placed in a high-pressure vessel, dipped in a thermostat. CO_2_ was pumped into the vessel using a plunger pump (HKS-12000, HY Accuracy, Seoul, Republic of Korea). The sorption pressure was maintained at 10 MPa, and the saturation temperatures varied over a range of −20 to 40 °C. After the sorption process, the sample was taken out from the vessel by fast depressurization (~1 s) and was immediately weighed. The concentration of CO_2_ in the polymer matrix was calculated as $C=\left(m-m_{0}\right)/m_{0}$ where C is the concentration of CO_2_ in the polymer (mg/g), m is the weight of sample after sorption, and $m_{0}$ is the initial weight of the sample. Different sorption durations were performed until no further weight increase was observed. This procedure was repeated at least three times for each condition. Fick’s second law was used to determine the saturated concentration of CO_2_ and the diffusion coefficient of CO_2_ in the polymer matrix. Since the diameter of the disc was larger than the depth by a factor of 20, the following unidirectional diffusion equation was used to estimate the diffusion coefficient and the saturation concentration [16].
(1)$$\frac{C}{C_{sat}}=1-\overset{\infty }{\underset{n=0}{\sum }}\frac{8}{{\left(2n+1\right)}^{2}\pi^{2}}exp\left(-\frac{D{\left(2n+1\right)}^{2}\pi^{2}t}{4l^{2}}\right)$$
here, t is the sorption time, l is the depth of the disc, C is the concentration of CO_2_ at time t, $C_{sat}$ is the saturated CO_2_ concentration in ABS/R-PET blend and D is the diffusion coefficient of CO_2_. $C_{sat}$ and D were obtained by fitting the equation to the experimental data. 

### 2.4. Preparation of ABS/R-PET Foam Structure

A two-step foaming process was conducted in order to prepare the ABS/R-PET microcellular foams. In the first step, a disc-type specimen was placed in a vessel, and CO_2_ was delivered into the vessel. The specimen was fully saturated with CO_2_ at the pressure of 10 MPa and the temperature of −20 °C for a period of more than 24 h. The depressurization rate was controlled to be either fast (~10 MPa/1 s) or slow (~10 MPa/3 min). After the first step, a thermal contrast method was used for cell nucleation and growth. The sample obtained after depressurization was immersed in either a hot water (under 100 °C) or ethylene glycol/water (over 100 °C) medium for 3 min. The heating bath temperature ranged from 60 °C to 120 °C. Then the sample was quenched at −20 °C in 50:50 v/v % aqueous ethanol solution for 5 min and dried at room temperature.

### 2.5. Structural Characterization

Field-emission scanning electron microscopy (FE-SEM, SUPRA 55VP, Carl Zeiss, Oberkochen, Germany) was used to observe the structural characteristics of the disc and the foams. The morphology of raw ABS/R-PET blend was observed following the method used by Cook et al. [6]. In this method, the ABS/R-PET blend was etched with 10 wt % of KOH ethanol solution for 20 h. This procedure removes the R-PET domain selectively from the blend. To observe the interior structure of foams, the samples were frozen in liquid nitrogen for a minute and fractured. The surface of the fractured specimen was coated with platinum using the sputtering coater (SCD005, BAL-TEC, Los Angeles, CA, US) and analyzed by FE-SEM. ImageJ [17] and the MorpholibJ library [18] were used to analyze the SEM images. The cell diameters were calculated as $d_{cell}=\sqrt{4A/\pi }$, where $d_{cell}$ is the diameter of the cell and A is the area of cell. At least 400 cells for each sample were measured in order to obtain the cell size distribution. Cell number density, which is defined as the number of cells per the volume of the unprocessed specimen, was calculated by the following procedure [13]. First, a specific area (A) in a SEM image was randomly chosen. Then, the number of cells (N) in the area was counted. These values were incorporated to calculate the cell number density ($N_{cell}$):(2)$$N_{cell}={\left(\frac{NM^{2}}{A}\right)}^{\frac{3}{2}}\left(\frac{\rho_{0}}{\rho }\right)$$
where M is the magnification factor, $\rho_{0}$ is the apparent density of the raw material, and ρ is the apparent density of the foam. The apparent densities of the sample were estimated by measuring the weight and volume. The weight of each specimen was measured by an analytical scale (EPG214, OHAUS), and the volume was determined by the amount of water displacement when the sample was immersed in water [19,20].

## 3. Results and Discussions

### 3.1. Characterization of Unprocessed ABS/R-PET Blend

Figure 1 shows the structure of the ABS/R-PET blend treated by a KOH/ethanol solution. Small pores were dispersed in the continuous phase. This means that small PET domains were dispersed in ABS terpolymer since the KOH solution hydrolyzed PET selectively. Hence, similar to other immiscible polymer blends, the ABS/R-PET blend also had the sea-island morphology [21,22]. This heterogeneous structure had different sorption or foaming behavior in each polymer region [22,23].

### 3.2. Measurement of Sorption Kinetics

Figure 2 shows the sorption behavior of CO_2_ in ABS/R-PET disc at 10 MPa. The concentration of CO_2_ steeply increased initially, and slowly converged to the saturation concentration. The curves were well fitted to Equation (1). The sorption amount and the time that it took to reach the saturation concentration, reached a maximum at −20 °C and decreased as the temperature increased. The CO_2_ uptake was reduced to approximately 65 % as the temperature increased to 40 °C, while the sorption time decreased to 2 h. The CO_2_ uptake and diffusivity at −20 °C were 175.3 mg/g and 0.516 × 10^−7^ cm^2^/s (Table 1), which were lower than those of ABS ($C_{\mathrm{sat}}=282.2\;\mathrm{mg}/g$ and D = 0.228 × 10^−6^ cm^2^/s) [10]. A decrease in CO_2_ uptake and diffusivity can result from either the CO_2_-induced crystallization of PET or the poor miscibility between PET and CO_2_. The sorption curves (Figure 2) and the DSC thermogram (Figure 3) of the polymer blend demonstrate that no CO_2_-induced crystallization of PET occurred in the ABS/R-PET blend. When the CO_2_-induced crystallization occurs in a neat PET sample, a “knee-like” curve [11] is observed because the absorbed CO_2_ in PET is excluded from the polymer matrix after the crystallization. This behavior was not observed in the polymeric blend. In addition, the thermogram of the microcellular foams did not significantly change compared to the unprocessed sample (Figure 3). Thus, no considerable CO_2_-induced crystallization occurred since the ABS terpolymer prevents the chain movement of R-PET polymer as Zhao et al. observed [12].

Thus, low CO_2_ uptake and diffusivity in the polymeric blend resulted from the weak PET/CO_2_ interaction. The Hansen solubility parameter (HSP) can be utilized to compare the extent of interaction between polymers and CO_2_ [24]. The HSPs of polymers were obtained from Hansen’s work [25] and those of CO_2_ at −20 °C and 10 MPa were calculated based on the Williams method [26]. Since the absorption of CO_2_ in the ABS matrix was largely dependent on the polybutadiene (PB) portion [9], the preference of CO_2_ was examined through the comparison of the Hansen distance ($R_{a}$) of PET and PB. It was hypothesized that the HSPs of polymers do not vary significantly from those at the ambient condition since the volumetric properties of polymeric materials do not largely depend on the pressure. The Hansen distance was determined by the following equation [27]:(3)$$R_{a}^{2}=4{\left(\delta_{d,2}-\delta_{d,1}\right)}^{2}+{\left(\delta_{p,2}-\delta_{p,1}\right)}^{2}+{\left(\delta_{h,2}-\delta_{h,1}\right)}^{2}$$
where $\delta_{d}$ is the energy from the dispersion force of the component, $\delta_{p}$ is the energy from the dipolar intermolecular force of the component, and $\delta_{h}$ is the energy from the hydrogen bond of the component. The Hansen distance between CO_2_ and PB was closer than that between CO_2_ and PET (Table 2). It means that CO_2_ was favorably absorbed in ABS domains. Thus, the existence of PET in the blend caused a decrease in CO_2_ uptake and diffusivity in the blend.

### 3.3. Characterization of the ABS/R-PET Microcellular Foams

Figure 4 shows the SEM images and the cell size distributions of ABS/R-PET foams. As the foaming temperature ($T_{f}$) increased, larger cells were more frequently observed and the cell size distributions broadened. This result was consistent with that of ABS polymeric foams, which showed an increase of the mean cell diameter as the temperature changed from 50 to 120 °C [9]. The mean cell diameter slightly increased from 1068 nm to 1195 nm. This slight increase of the cell diameter can be explained as follows: When $T_{f}$ was low, the cell growth was limited by the rigidity of the polymer matrix. At high $T_{f}$, the growth rate of the CO_2_ bubble increased and the polymer matrix became less viscous by an active chain movement. Thus, the expansion of CO_2_ can push the polymeric wall enveloping the bubble, which facilitates the cell growth [28].

On the other hand, the standard deviation of cell diameters at $T_{f}=60\;{}^{\circ}C$ was much smaller than those at higher $T_{f}$ s (Table 3). This reflects the structural heterogeneity of the polymer blend. As mentioned above, the butadiene portion of the ABS terpolymer was more CO_2_-philic than PET. The $T_{g}$ of an ABS terpolymer drastically decreased under a high-pressure CO_2_ environment. Furthermore, the ABS terpolymer was rubbery at any sorption temperatures when the sorption pressure is higher than 5 MPa [8]. In contrast, PET has a low CO_2_ uptake, and the extent of the $T_{g}$ depression was relatively low. The $T_{g}$ of PET under the high-pressure CO_2_ environment decreased to approximately 40 °C [29]. Hence, the state of each ABS and PET polymer was rubbery and glassy respectively when the blend was saturated with CO_2_ at −20 °C, 10 MPa. This heterogeneous structure of the polymer blend led to a formation of small cells in the ABS domain at $T_{f}=60\;℃$. The foams should be only created in the amorphous ABS region, since the PET region was glassy. Also, the presence of a glassy PET domain resulted in relatively uniform cells with a mean diameter of ~1 μm since they restricted the cell growth. When $T_{f}$ was 80 °C, PET domain also became rubbery, which resulted in the formation of cells in both domains. Consequently, an increase of the cell diameter and broadening of the size distribution at high $T_{f}$ came from the increase of the extent of cell growth and the heterogeneous structure of the polymeric blend. The obtained foam structure which has both large and small cells, can improve both the thermal and the mechanical properties [30].

Next, the effect of the depressurization rate was examined. When the thermal contrast method was not applied, both samples after fast and slow depressurization did not show any distinct differences (Figure 5a,b). The cell number densities of the foams with a simple depressurization step (Figure 5a,b) were smaller than those from the additional thermal contrast method (Figure 5c,d). In contrast to the samples without thermal contrast, the foams after the thermal contrast method with fast and slow depressurization rate were different from each other. The mean cell diameter did not change significantly, but the cell number density of the sample from fast depressurization was larger than that from slow depressurization, by a factor of 2.7 (Table 4). This result suggests that the depressurization does not affect cell growth, but cell nucleation. The remaining CO_2_ concentration after fast depressurization was 175.3 mg/g, which was higher than that after slow depressurization (154.3 mg/g). A sudden pressure drop induced more CO_2_ to remain in the polymer. Thus, rapid depressurization led to the formation of a large number of cell nuclei, due to the high extent of supersaturation [31]. When the thermal contrast was not applied, these cell nuclei could not grow, since CO_2_ slowly diffuses out from the polymer matrix. Hence, the effect of the depressurization rate on the bubble nucleation cannot be observed well in Figure 5a,b. On the contrary, dipping the samples into a high-temperature medium allowed the nuclei to grow.

## 4. Conclusions

In conclusion, this work demonstrated that a microcellular foam of the ABS/R-PET blend can be successfully prepared using subcritical CO_2_ as a blowing agent. The microcellular foams had a mean cell size of 1 μm, and the size distribution largely depended on the foaming temperature. As the foaming temperature increased, larger cells were observed because both ABS and R-PET became rubbery by CO_2_-induced plasticization. The depressurization rate largely influenced the extent of the cell nucleation. Hence, a fast depressurization rate and the large temperature difference between the saturation temperature and the foaming temperature are desirable in the preparation of ABS/R-PET microcellular foams. Considering that a large amount of the post-consumer PET is used as a constituent of a polymer blend, this work will be helpful for the extended utilization of recycled PET.

## Figures and Tables

**Figure 1 polymers-11-00291-f001:**
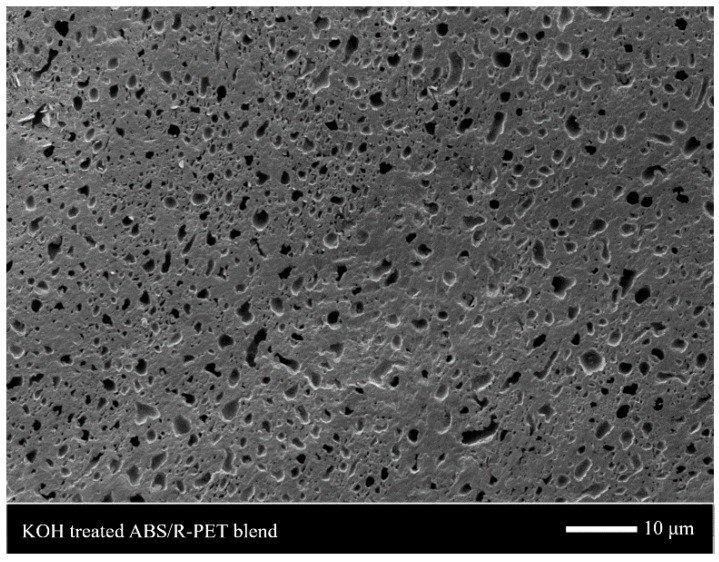
The SEM image of ABS/R-PET (acrylonitrile-butadiene-styrene/recycled polyethylene terephthalate) blend treated with KOH/ethanol solution. The small pores that were dispersed in the ABS/R-PET domain were formed by etching the R-PET.

**Figure 2 polymers-11-00291-f002:**
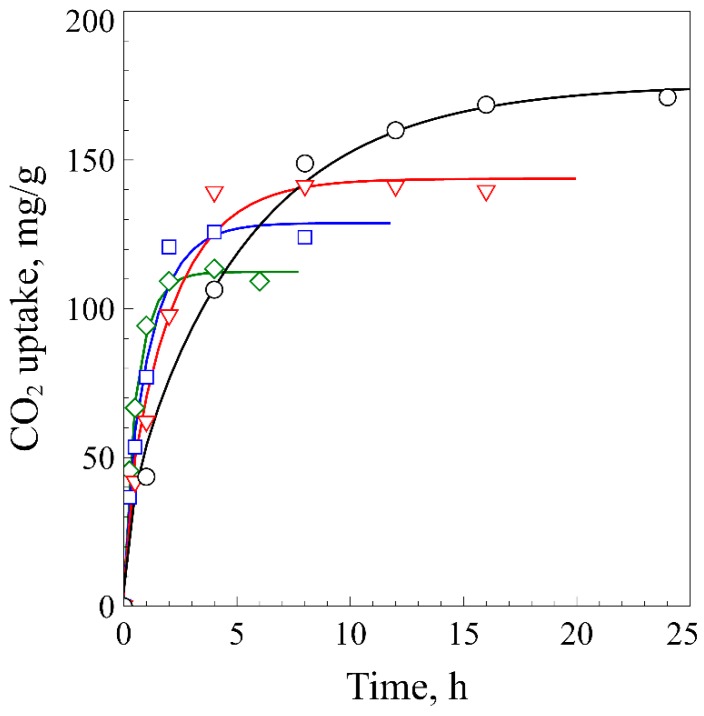
The CO_2_ sorption kinetics of ABS/R-PET at 10 MPa and the temperature of −20 °C (Circle), 0 °C (Triangle), 20 °C (Square), and 40 °C (Diamond). The experimental data were fitted to Equation (2). The amount of CO_2_ absorbed in the blend increased as the saturation temperature decreased, whereas the time to reach the saturation became longer.

**Figure 3 polymers-11-00291-f003:**
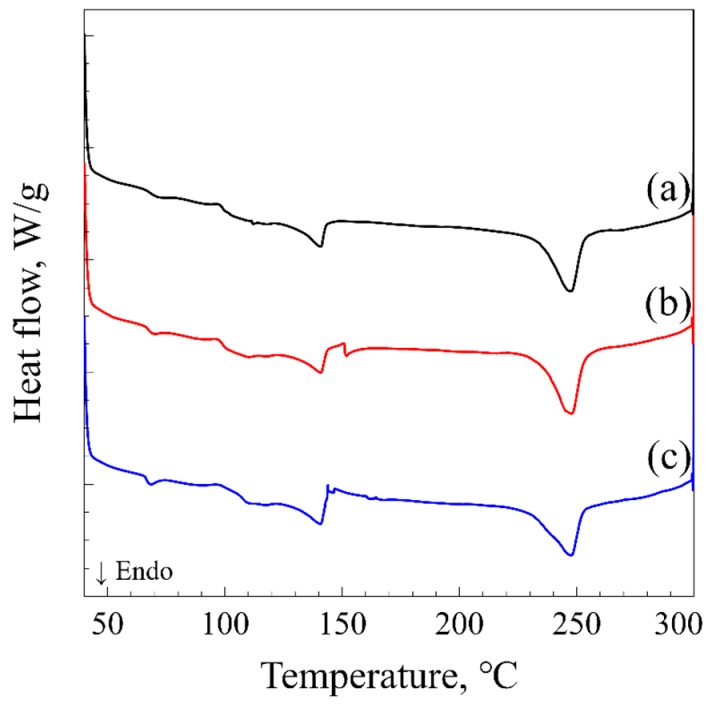
Differential scanning calorimetry (DSC) thermogram of (**a**) unprocessed ABS/R-PET blend, (**b**) ABS/R-PET blend after sorption process, and (**c**) ABS/R-PET blend foamed at 60 °C. The exothermic peak near 150 °C is the cold crystallization peak of PET, and the endothermic peak at 250 °C is the melting peak of PET.

**Figure 4 polymers-11-00291-f004:**
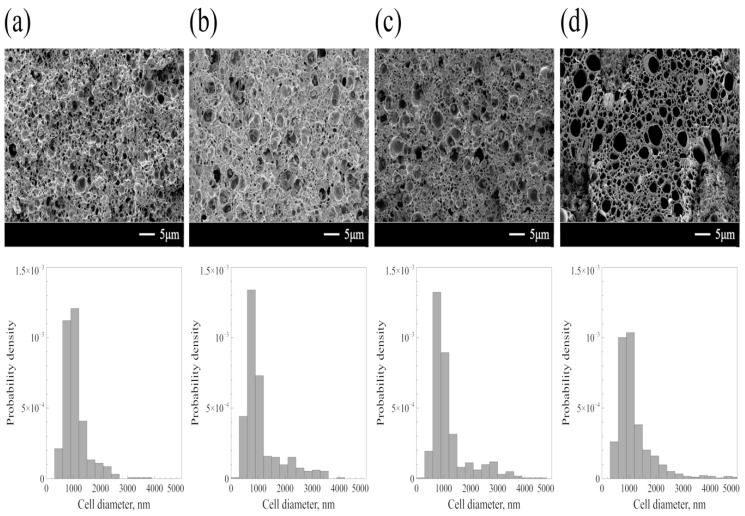
SEM images and the size distribution curves of ABS/R-PET blend foams. Samples were foamed at (**a**) 60 °C, (**b**) 80 °C, (**c**) 100 °C, and (**d**) 120 °C after sorption with CO_2_ at −20 °C, 10 MPa over a period of 24 h. The depressurization time was one second. The size distribution curves became broader and long-tailed as the $T_{f}$ increased.

**Figure 5 polymers-11-00291-f005:**
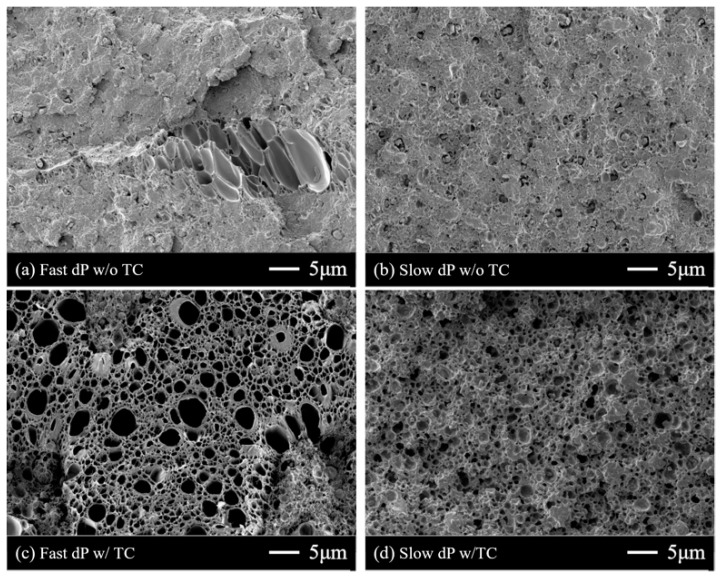
SEM images of processed ABS/PET blends after (**a**) depressurization for 1 s, (**b**) depressurization for 3 min, (**c**) depressurization for 1 s and thermal contrast at 120 °C, and (**d**) depressurization for 3 min and thermal contrast at 120 °C. When the thermal contrast was performed after depressurization, a large number of cells appeared.

**Table 1 polymers-11-00291-t001:** The saturation concentrations (C_sat_) and diffusion coefficients (D) of CO_2_ in ABS/R-PET blend.

Saturation Temperature, °C	C_sat_, mg/g ^a^	D, 10^−7^ cm^2^/s ^a^
−20	175.3 (14.25)	0.516 (0.17)
0	143.7 (11.80)	1.304 (0.47)
20	128.8 (15.75)	2.264 (1.03)
40	112.4 (5.40)	4.083 (0.85)

^a^ indicates the uncertainties of the estimation with 95% confidence bounds were presented in parentheses.

**Table 2 polymers-11-00291-t002:** The Hansen solubility parameters (HSPs) of CO_2_ and polymers and the Hansen distance from CO_2_.

Component	$\delta_{d},\;{\left(MPa\right)}^{1/2}$	$\delta_{p},\;{\left(\mathrm{MPa}\right)}^{1/2}$	$\delta_{h},\;{\left(\mathrm{MPa}\right)}^{1/2}$	$R_{a},\;{\left(\mathrm{MPa}\right)}^{1/2}$
CO_2_ (−20 °C, 10 MPa)	14.6	5.1	6.0	-
PET	18.7	6.3	6.7	8.3
PB	17.3	2.3	3.4	6.6

**Table 3 polymers-11-00291-t003:** The mean cell diameter and standard deviation of ABS/R-PET blend foams.

$T_{f},\;{}^{\circ}C$	Mean Cell Diameter, nm	Standard Deviation, nm
60	1068	445
80	1118	686
100	1192	717
120	1194	701

**Table 4 polymers-11-00291-t004:** The properties of the ABS/R-PET blends foamed at 120 °C after fast or slow depressurization.

Depressurization Rate	Mean Cell Diameter, nm	Sample Density, g/cm^3^	Cell Number Density, 10^12^ cells/cm^3^	The Remaining CO_2_, mg/g ^a^
Fast	1195	0.36	1.86	175.3
Slow	1060	0.40	0.69	154.3

^a^ indicates the remaining CO_2_ concentration before thermal contrast process.

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
