# Peer review of "Solid-State Foaming of Acrylonitrile-Butadiene-Styrene/Recycled Polyethylene Terephthalate Using Carbon Dioxide as a Blowing Agent"

_polymers, 2019, doi:10.3390/polym11020291_

Round 1

Reviewer 1 Report

The manuscript entitled "Solid-state foaming of acrylonitrile-butadiene-styrene / recycled polyethylene terephthalate using carbon dioxide as a blowing agent" is a work dealing with the preparation of foams from ABS/PET blends with the use of supercritical CO2. This method is well established and there is a lot of research currently done with the use of a variety of different polymers and parameters.

The manuscript although not of high novelty can contribute to the topic of the foams, therefore, I suggest some revision to the following topics and then considering again for acceptance.

- Was it possible to perform pressure depended on DSC experiments or DSC experiment in the presence of CO2 instead of Nitrogen to monitor the effect of plasticization?

- The high standard deviation shows that the foam structure of the foam is not homogeneous and that there are big and small cells everywhere. Is it good for the prepared porous material and for its applications? Can the authors consider a method for controlling the porosity?

Author Response

The authors are grateful to the reviewer for the constructive criticisms. The insightful comments helped the authors improve the manuscript. The detailed responses to all the comments are as follows.

Point 1: Was it possible to perform pressure depended on DSC experiments or DSC experiment in the presence of CO2 instead of Nitrogen to monitor the effect of plasticization?

Response 1: Plasticization of each polymer (ABS and PET) at CO2 environment was presented in reference 8 and 29, respectively. Using DSC experiment, they measured a decrease of glass transition temperature when the pressure of CO2 increased. The related contents were described in introduction (Page 2, 48-49) and section 3.3 (Page 8, 227-231).

Point 2: The high standard deviation shows that the foam structure of the foam is not homogeneous and that there are big and small cells everywhere. Is it good for the prepared porous material and for its applications? Can the authors consider a method for controlling the porosity?

Response 2:

1 Is it good for the prepared porous material and for its applications?

 : The presence of big and small cells in the porous structure offers excellent properties since low thermal conductivity constant resulted from big cells, and improved mechanical properties resulted from small cells. The sentence and related reference were added to the section 3.3 (Page 8, 240-241).

2 Can the authors consider a method for controlling the porosity?

 : The various porous structures can be prepared when the foaming process conditions, such as foaming temperature, were varied. The detailed sentence was described in the section 3.3 (Page 7, 213-221) and conclusion (Page 9, 269-279).

Reviewer 2 Report

In this manuscript the authors prepared a microcellular foam of ABS/R-PET blend with subcritical CO2 as a blowing agent. This work will be helpful for the extended utilization of the recycled PET, it supply some valuable information for the related readers. Considering the present form, it can be accepted in Polymers.

Author Response

Response: The authors appreciate the time the reviewer took to evaluate the manuscript. The authors are grateful for the comments.